# Flow Induced Symmetry Breaking in a Conceptual Polarity Model

**DOI:** 10.3390/cells9061524

**Published:** 2020-06-23

**Authors:** Manon C. Wigbers, Fridtjof Brauns, Ching Yee Leung, Erwin Frey

**Affiliations:** Arnold Sommerfeld Center for Theoretical Physics (ASC) and Center for NanoScience (CeNS), Department of Physics, Ludwig-Maximilians-Universität München, Theresienstraße 37, D–80333 München, Germany; m.wigbers@physik.uni-muenchen.de (M.C.W.); fridtjof.brauns@physik.uni-muenchen.de (F.B.); c.leung@physik.uni-muenchen.de (C.Y.L.)

**Keywords:** symmetry breaking, cytoplasmic flow, phase-space analysis, pattern formation

## Abstract

Important cellular processes, such as cell motility and cell division, are coordinated by cell polarity, which is determined by the non-uniform distribution of certain proteins. Such protein patterns form via an interplay of protein reactions and protein transport. Since Turing’s seminal work, the formation of protein patterns resulting from the interplay between reactions and diffusive transport has been widely studied. Over the last few years, increasing evidence shows that also advective transport, resulting from cytosolic and cortical flows, is present in many cells. However, it remains unclear how and whether these flows contribute to protein-pattern formation. To address this question, we use a minimal model that conserves the total protein mass to characterize the effects of cytosolic flow on pattern formation. Combining a linear stability analysis with numerical simulations, we find that membrane-bound protein patterns propagate against the direction of cytoplasmic flow with a speed that is maximal for intermediate flow speed. We show that the mechanism underlying this pattern propagation relies on a higher protein influx on the upstream side of the pattern compared to the downstream side. Furthermore, we find that cytosolic flow can change the membrane pattern qualitatively from a peak pattern to a mesa pattern. Finally, our study shows that a non-uniform flow profile can induce pattern formation by triggering a regional lateral instability.

## 1. Introduction

Many biological processes rely on the spatiotemporal organization of proteins. Arguably one of the most elementary forms of such organization is cell polarization—the formation of a “cap” or spot of high protein concentration that determines a direction. Such a polarity axis then coordinates downstream processes including motility [1,2], cell division [3], and directional growth [4]. Cell polarization is an example for symmetry breaking [5], as the orientational symmetry of the initially homogeneous protein distribution is broken by the formation of the polar cap.

Intracellular protein patterns arise from the interplay between protein interactions (chemical reactions) and protein transport. Diffusion in the cytosol serves as the most elementary means of transport. Pattern formation resulting from the interplay of reactions and diffusion has been widely studied since Turing’s seminal work [6]. In addition to diffusion, proteins can be transported by fluid flows in the cytoplasm [7,8,9] and along cytoskeletal structures (vesicle trafficking, cortical contractions) driven by molecular motors [10,11,12]. These processes lead to advective transport of proteins.

Recently, it has been shown experimentally that advective transport (caused by cortical flows) induces polarization of the PAR system in the *C. elegans* embryo [13,14,15]. Furthermore, in vitro studies with the MinDE system of *E. coli*, reconstituted in microfluidic chambers, have shown that the flow of the bulk fluid has a strong effect on the protein patterns that form on the membrane [16,17]. Increasing evidence shows that cortical and cytosolic flows (also called “cytoplasmic streaming”) are present in many cells [18,19,20,21,22,23]. In addition, cortical contractions can drive cell-shape deformations [24], inducing flows in the incompressible cytosol [8,25]. However, the role of flows for protein-pattern formation remains elusive. This motivates to study the role of advective flow from a conceptual perspective, with a minimal model. The insights thus gained will help to understand the basic, principal effects of advective flow on pattern formation and reveal the underlying elementary mechanisms.

The basis of our study is a paradigmatic class of models for cell polarization that describe a single protein species which has a membrane-bound state and a cytosolic state. Such two-component mass-conserving reaction–diffusion (2cMcRD) systems serve as conceptual models for cell polarization [7,26,27,28,29,30,31]. Specifically, they have been used to model Cdc42 polarization in budding yeast [32] and PAR-protein polarity [33]. 2cMcRD systems generically exhibit both spontaneous and stimulus-induced polarization [5,31,33]. In the former case, a spatially uniform steady state is unstable against small spatial perturbations (“Turing instability” [6]). Adjacent to the parameter regime of this lateral instability, a sufficiently strong, localized stimulus (e.g., an external signal) can induce the formation of a pattern starting from a stable spatially uniform state. The steady state patterns that form in two-component McRD systems are generally stationary (there are no traveling or standing waves). Moreover, the final stationary pattern has no characteristic wavelength. Instead, the peaks that grow initially from the fastest growing mode (“most unstable wavelength”) compete for mass until only a single peak remains (“winner takes all”) [30,34,35]. The location of this peak can be controlled by external stimuli (e.g., spatial gradients in the reaction rates) [34,36].

Recently, a theoretical framework, termed local equilibria theory, has been developed to study these phenomena using a geometric analysis in the phase plane of the protein concentrations [31,37]. With this framework one can gain insight into the mechanisms underlying the dynamics of McRD systems both in the linear and in the strongly nonlinear regime, thereby bridging the gap between these two regimes.

Here, we show that cytosolic flow in two-component systems always induces upstream propagation of the membrane-bound pattern. In other words, the peak moves against the cytosolic flow direction. This propagation is driven by a higher protein influx on the upstream side of the membrane-concentration peak compared to its downstream side. Using this insight, we are able to explain why the propagation speed becomes maximal at intermediate flow speeds and vanishes when the rate of advective transport becomes fast compared to the rate of diffusive transport or compared to the reaction rates. We first study a uniform flow profile using periodic boundaries. This effectively represents a circular flow, which is observed in plant cells (where this phenomenon is called cytoplasmic streaming or cyclosis) [38]. It also represents an in vitro system in a laterally large microfluidic chamber. We then study the effect of a spatially non-uniform flow profile in a system with reflective boundaries, as a minimal system for flows close to the membrane [7,13,15], e.g., in the actin cortex. We show that a non-uniform flow profile redistributes the protein mass, which can trigger a regional lateral instability and thereby induce pattern formation from a stable homogeneous steady state.

The remainder of the paper is structured as follows. We first introduce the model in Section 2. We then perform a linear stability analysis in Section 3 to show how spatially uniform cytosolic flow influences the dynamics close to a homogeneous steady state. In Section 4, we use numerical simulations to study the fully nonlinear long-term behavior of the system. Next, we show that upon increasing the cytosolic flow velocity, the pattern can qualitatively change from a mesa pattern to a peak pattern in Section 5. Finally, in Section 6, we study how a spatially non-uniform cytosolic flow can trigger a regional lateral instability and thus induce pattern formation. Implications of our findings and links to earlier literature are briefly discussed at the end of each section. We conclude with a brief outlook section.

## 2. Model

We consider a spatially one-dimensional system of length *L*. The proteins can cycle between a membrane-bound state (concentration m(x,t)) and a cytosolic state (concentration c(x,t)), and diffuse with diffusion constants $D_{m}$ and $D_{c}$, respectively (Figure 1). In cells, the diffusion constant on the membrane is typically much smaller than the diffusion constant in the cytosol. In the cytosol, the proteins are assumed to be advected with a speed $v_{f}\left(x\right)$, as indicated by the blue arrow in Figure 1. Thus, the reaction-diffusion-advection equations for the cytosolic density and membrane density read
(1a)$$\begin{array}{cc} \partial_{t}c+\partial_{x}\left(v_{f}c\right) & =\;D_{c}\partial_{x}^{2}c\;-f\left(m,c\right), \end{array}$$
(1b)$$\begin{array}{cc} \partial_{t}m & =D_{m}\partial_{x}^{2}m+f\left(m,c\right), \end{array}$$
with either periodic or reflective boundary conditions. The nonlinear function f(m,c) describes the reaction kinetics of the system. Attachment–detachment kinetics can generically be written in the form
(2)f(m,c)=a(m)c−d(m)m,
where a(m)>0 and d(m)>0 denote the rate of attachment from the cytosol to the membrane and detachment from the membrane to the cytosol, respectively. The dynamics given by Equation (1) conserve the average total density
(3)$$\bar{n}=\frac{1}{L}\int_{0}^{L}\;dx\;n\left(x,t\right).$$

Here, we introduced the local total density n(x,t):=m(x,t)+c(x,t).

For illustration purposes, we will use a specific realization of the reaction kinetics [31],
(4)$$a\left(m\right)=k_{\mathrm{on}}+k_{\mathrm{fb}}m\;\mathrm{and}\;d\left(m\right)=\frac{k_{\mathrm{off}}}{K_{D}+m},$$
describing attachment with a rate $k_{\mathrm{on}}$, self-recruitment with a rate $k_{\mathrm{fb}}$, and enzyme-driven detachment with a rate $k_{\mathrm{off}}$ and the Michaelis–Menten constant $K_{D}$, respectively. However, our results do not depend on the specific choice of the reaction kinetics. Unless stated otherwise, we use the parameters: $k_{\mathrm{on}}=1\;s^{-1},k_{\mathrm{fb}}=1\;\mu m\;s^{-1},k_{\mathrm{off}}=2\;s^{-1},K_{D}=1\;\mu m^{-1},\bar{n}=5\;\mu m^{-1},D_{m}=0.01\;\mu m^{2}\;/s,D_{c}=10\;\mu m^{2}\;/s$.

## 3. Linear Stability Analysis

### 3.1. Linearized Dynamics and Basic Results

To study how cytosolic flow affects the formation of protein patterns, we first consider a spatially uniform flow profile (i.e., constant $v_{f}\left(x\right)=v_{f}$) and perform a linear stability analysis of a spatially homogeneous steady state $u^{*}=\left(c^{*},m^{*}\right)$:(5)$$f\left(m^{*},c^{*}\right)=0,\;m^{*}+c^{*}=\bar{n}.$$

Following the standard procedure, we linearize the dynamics for small perturbations $u\left(x,t\right)=\left(c(x,t),m(x,t)\right)=u^{*}+\delta u\left(x,t\right)$ around the homogeneous steady state. Expanding δu(x,t) in exponentially growing (or decaying) Fourier modes $\delta u={\hat{u}}_{q}\;e^{\sigma t}e^{iqx}$ leads to the eigenvalue problem
(6)$$J{\hat{u}}_{q}=\sigma {\hat{u}}_{q},$$
with the Jacobian
$$J=\left(\begin{array}{cc} -D_{c}q^{2}-iv_{f}q-f_{c} & -f_{m}\\ f_{c} & -D_{m}q^{2}+f_{m} \end{array}\right),$$
where $f_{c}=\partial_{c}{f|}_{u^{*}}$ and $f_{m}=\partial_{m}{f|}_{u^{*}}$ encode the linearized reaction kinetics. Note that for reaction kinetics of the form Equation (Equation 2), $f_{c}=a\left(m\right)>0$ and we consider this case in the following.

For each mode with wavenumber *q*, there are two eigenvalues $\sigma_{1,2}\left(q\right)$. The case q=0 corresponds to spatially homogeneous perturbations, where the two eigenvalues are given by $\sigma_{1}=f_{m}-f_{c}$ and $\sigma_{2}=0$ [31]. Here, we restrict our analysis to homogeneously stable states ($\sigma_{1}<0$). The second eigenvalue ($\sigma_{2}=0$) corresponds to perturbations that change the average mass $\bar{n}$ and therefore shift the homogeneous steady state $u^{*}\left(\bar{n}\right)$ along the nullcline f=0. As a result that these perturbations break mass-conservation, they are not relevant for the stability of a closed system as considered here. The modes q>0 determine the stability of the system against spatially inhomogeneous perturbations (lateral stability). The eigenvalue with the larger real part determines the stability and will be denoted by σ(q), suppressing the index.

A typical dispersion relation with a band of unstable modes is shown in Figure 2A. The real part (solid line), indicating the mode’s growth rate, has a band of unstable modes $\left[0,q_{\max}\right]$ where Reσ(q)>0. The fastest growing mode $q^{*}$ determines the wavelength λ of the pattern that initially grows, triggered by a small, random perturbation of the spatially homogeneous steady state. For $v_{f}=0$, the imaginary part of σ(q) vanishes, for locally stable steady states (σ(0)≤0) [31]. However, in the presence of flow, the imaginary part of σ(q) is non-zero (dashed line in Figure 2A), which implies a propagation of each mode with the phase velocity vphase(q)=−Imσ(q)/q. This means that a mode *q* not only grows over time (orange arrows in Figure 2B), but also propagates as indicated by the pink arrows in Figure 2B. Further below, in Section 3.4, we will show that Imσ(q) always has the same sign as the flow velocity $v_{f}$, such that all modes propagate against the flow direction.

To gain physical insight into the mechanisms underlying the growth and propagation of perturbations (modes) we will first give an intuitive explanation of a lateral instability in McRD systems, building on the concepts of local equilibria theory [31,37]. We then provide a more detailed analysis in the limits of long wavelength as well as fast and slow flow.

### 3.2. Intuition for the Flow-Driven Instability and Upstream Propagation of the Unstable Mode

Lateral instability in McRD systems can be understood as a mass-redistribution instability [31]. Let us briefly recap the mechanism underlying this instability for a system without flow. To this end, we first discuss the effect of reactions and diffusion separately, and explain how these effects together drive the mass-redistribution instability. We then explain how this instability is affected by cytosolic flow.

Consider a spatially homogeneous steady state, perturbed by a slight redistribution of the local total density n(x,t). The dashed orange line in Figure 3A shows such a perturbation where the membrane concentration (Figure 3A top) is slightly perturbed in a sinusoidal fashion. In phase space this is represented by a density distribution that slightly deviates from the spatially homogeneous steady state (marked by the orange dashed line). Here, the open star and open circle mark the minimum and maximum of the local total density, respectively. The local total density determines the local reactive equilibrium concentrations $m^{*}\left(n\right)$ and $c^{*}\left(n\right)$ (cf. Equation (Equation 5), replacing the average mass $\bar{n}$ by the local mass n(x,t)). In phase space (Figure 3A bottom) these local equilibria can be read off from the intersections (marked by black circles) of the reactive subspaces n(x,t)=m(x,t)+c(x,t) (gray solid lines) and the reactive nullcline (black solid lines). A slight redistribution of the local total density shifts the reactive equilibria, leading to reactive flows towards these shifted equilibria (red and green arrows in Figure 3A). Thus, the reactive equilibria, and thereby the reactive flows, are encoded in the shape of the reactive nullcline in phase space. If the nullcline slope is negative, increasing the total density leads to a decreasing equilibrium cytosolic concentration and therefore to attachment (green arrows in Figure 3A). Conversely, in regions of lower total density, the equilibrium cytosolic concentration increases via detachment (red arrows in Figure 3A). Hence, regions of high total density become self-organized attachment zones and regions of low total density become self-organized detachment zones [37] (green and red areas in Figure 3 top and middle).

These attachment and detachment zones act as sinks and sources for diffusive mass-transport on the membrane and in the cytosol: The attachment zone acts as a cytosolic sink and membrane source, and the detachment zone acts as a cytosolic source and a membrane sink (blue arrows in Figure 3B). As diffusion in the cytosol is much faster than in the membrane, mass is transported faster in the cytosol than on the membrane, as indicated by the size of the blue arrows in Figure 3B top and middle. This leads to net mass transport from the detachment zone to the attachment zone. As the local total density increases in the attachment zone, it facilitates further attachment and thereby the growth of the pattern on the membrane. In short, the mechanism underlying the mass-redistribution instability is a cascade of attachment–detachment kinetics (Figure 3A) and net mass-transport towards attachment zones (Figure 3B).

How does cytosolic fluid flow affect the mass-redistribution instability? Cytosolic flow transports proteins advectively. This advective transport shifts the cytosolic density profile downstream relative to the membrane density profile (dashed to solid orange line in Figure 3C middle). This shift leads to an increase of the cytosolic density on the upstream (cyan) side of the membrane peak and a decrease on the downstream (magenta) side, in Figure 3C (middle), respectively. In phase space, this asymmetry is reflected as a ‘loop’ shape of the phase space trajectory that corresponds to the real space pattern (Figure 3C bottom). The higher cytosolic density on the upstream side increases attachment relative to the downstream side. This leads to a propagation of the membrane concentration profile in the upstream direction.

### 3.3. Long Wavelength Limit

To complement this intuitive picture we consider the long wavelength limit q→0. (In principle, the dispersion relation can be easily obtained in closed form using the formula for eigenvalues of 2×2 matrices: $\sigma_{1,2}=\frac{1}{2}\mathrm{tr}J\mp \frac{1}{2}\sqrt{{\left(\mathrm{tr}J\right)}^{2}-4\det J}$ where trJ and detJ are the Jacobian’s trace and determinant, respectively. As a result that the resulting expression is rather lengthy, we do not write it out it explicitly here.) In this limit, the dispersion relation expanded to second order in *q* reads
(7)$$\sigma \left(q\right)\approx -\frac{1}{1+s_{\mathrm{nc}}}\left[is_{\mathrm{nc}}v_{f}q+\left(D_{m}+s_{\mathrm{nc}}D_{c}\right)q^{2}+\frac{s_{\mathrm{nc}}\;v_{f}^{2}}{f_{c}{\left(1+s_{\mathrm{nc}}\right)}^{2}}q^{2}\right],$$
where $s_{\mathrm{nc}}=-f_{m}/f_{c}$ is the slope of the reactive nullcline. The imaginary part Imσ(q) is linear in *q* to lowest order, implying a phase velocity $v_{\mathrm{phase}}=v_{f}s_{\mathrm{nc}}/\left(1+s_{\mathrm{nc}}\right)$ that is independent of the wavelength. The growth rate Reσ(q) is quadratic in *q* to lowest order. If this quadratic term is positive, there is a band of unstable modes (Homogeneous stability implies that the nullcline slope $s_{\mathrm{nc}}$ is larger than −1 [31], such that the prefactor ${\left(1+s_{\mathrm{nc}}\right)}^{-1}$ is positive.). Hence, the criterion for a mass-redistribution instability can be expressed in terms of the nullcline slope [31]
(8)$$s_{\mathrm{nc}}<-\frac{D_{m}}{D_{c}}{\left[1+\frac{v_{f}^{2}}{{\left(1+s_{\mathrm{nc}}\right)}^{2}D_{c}f_{c}}\right]}^{-1}\;.$$

In the absence of flow, $v_{f}=0$, we recover the slope criterion $s_{\mathrm{nc}}<-D_{m}/D_{c}$ for a mass-redistribution instability driven by cytosolic diffusion [31]. We find that flow always increases the range of instability since the second term in the square brackets monotonically increases with flow speed $|v_{f}|$. Furthermore, the instability criterion becomes independent of the diffusion constants in the limit of fast flow ($|v_{f}|\gg \sqrt{D_{c}f_{c}}$). The criterion for the (flow-driven) mass-redistribution instability then simply becomes $s_{\mathrm{nc}}<0$, independently of the ratio of the diffusion constants. This has the interesting consequence that, for sufficiently fast flow, a mass-redistribution instability can be driven solely via cytoplasmic flow, independent of diffusion.

### 3.4. Limits of Slow and fast Flow

To analyze the effect of flow for wavelengths away from the long wavelength limit, it is instructive to consider the limit cases of slow and fast flow speed.

We first consider a limit where advective transport ${\left(qv_{f}\right)}^{-1}$ is slow compared either to the chemical reactions or to diffusive transport. To lowest order in $v_{f}$, the dispersion relation is given by (see Appendix B)
(9)$$\sigma \left(q\right)\approx \sigma^{\left(0\right)}\left(q\right)+i\frac{v_{f}q}{2}A\left(q\right),$$
where the zeroth order term, $\sigma^{\left(0\right)}\left(q\right)$, is the dispersion relation in the absence of flow, which has no imaginary part [31] (cf. Equation (Equation 11)). The function A(q) is positive for all laterally unstable modes (Reσ(q)>0). Equation (Equation 9) shows that to lowest order (linear in $v_{f}$) the effect of cytosolic flow is to induce propagation of the modes with the phase velocity vphase(q)=−Imσ(q)/q≈−vfA(q). Since A(q)>0 for laterally unstable modes, all growing perturbations propagate against the direction of the flow (as illustrated in Figure 2B).

In the limit of fast flow (compared either to reactions or to cytosolic transport) we find that the dispersion relation (given by the eigenvalue problem Equation (Equation 6)) reduces to
(10)$$\sigma \left(q\right)\approx f_{m}-D_{m}q^{2}+i\frac{f_{c}f_{m}}{v_{f}q}$$
for non-zero wavenumbers. The real part of the dispersion relation in this fast flow limit becomes identical to the dispersion relation in the limit of fast diffusion [31]. In both limits, cytosolic transport becomes (near) instantaneous. In particular, in the limit of fast flow, advective transport completely dominates over diffusive transport in the cytosol such that the dispersion relation becomes independent of the cytosol diffusion constant $D_{c}$.

From the imaginary part of σ(q), we obtain the phase velocity $v_{\mathrm{phase}}=-f_{c}f_{m}/\left(v_{f}q^{2}\right)$. In other words, an increase in cytosolic flow leads to a decrease of the phase velocity. This is opposite to the slow flow limit discussed above, where the phase velocity increased linearly with the flow speed.

To rationalize these findings, we recall the propagation mechanism as discussed above. There, we argued that a phase shift between the membrane and the cytosol pattern is responsible for the pattern propagation, as it leads to an asymmetry in the attachment–detachment balance upstream and downstream. This phase shift increases with the flow velocity and eventually saturateslat π/4 (The phase shift can be read off from the real and imaginary parts of the eigenvectors in the linear stability analysis.). On the other hand, the cytosol concentration gradients become shallower the faster the flow. To understand why this is, imagine a small volume element in the cytosol being advected with the flow. The faster the flow, the less time it has to interact with each point on the membrane it passes. Therefore, for faster advective flow, the attachment–detachment flux at the membrane is effectively diluted over a larger cytosolic volume. This leads to a flattening of the cytosolic concentration profile (see Appendix A), and therefore a reduction in the upstream–downstream asymmetry of attachment. As a result, in the limit of fast flow, the pattern propagates slower the faster the flow, whereas, in the limit of slow flow, the pattern propagates faster the faster the flow. Thus, comparing these two limits, we learn that the phase velocity reaches a maximum at intermediate flow speeds.

### 3.5. Summary and Discussion of Linear Stability

Let us briefly summarize our main findings from linear stability analysis. We found that the leading order effect of cytosolic flow is to induce upstream propagation of patterns. This propagation is driven by the faster resupply of protein mass on the upstream side of the pattern compared to the downstream side. A similar effect was previously found for vegetation patterns which move uphill because nutrients are transported downhill by water flow [39]. Even though these systems are not strictly mass conserving, their pattern propagation underlies the same principle: The nutrient uptake in regions of high vegetation density creates a nutrient sink which is resupplied asymmetrically due to the downhill flow of water and nutrients.

Moreover, we used a phase-space analysis to explain how flow extends the range of parameters where patterns emerge spontaneously, i.e., where the homogeneous steady state is laterally unstable. This was previously shown mathematically for general two-component reaction–diffusion systems (not restricted to mass-conserving ones) [39,40]. Our analysis in the long wavelength limit explains the physical mechanism of this instability for mass-conserving systems: The flow-driven instability is a mass-redistribution instability, driven by a self-amplifying cascade of (flow-driven) mass transport and the self-organized formation of attachment and detachment zones (shifting reactive equilibria). This shows that the instability mechanism is identical to the mass-redistribution instability that underlies pattern formation in systems without flow (i.e., where only diffusion drives mass transport) [31]. For these systems, the instability strictly requires $D_{c}>D_{m}$. In contrast, we find that for sufficiently fast flow, there can be a mass-redistribution instability even in the absence of cytosolic diffusion ($D_{c}=0$). While the case $D_{c}=0$ is not physiologically relevant in the context of intracellular pattern formation, it may be relevant for the formation of vegetation patterns on sloped terrain [41], where *c* and *m* are the soil-nutrient concentration and plant biomass density, respectively. In conclusion, advective flow can fully replace diffusion as the mass-transport mechanism driving the mass-redistribution instability.

## 4. Pattern Propagation in the Nonlinear Regime

So far we have analyzed how cytosolic flow affects the dynamics of the system in the vicinity of a homogeneous steady state, using linear stability analysis. However, patterns generically do not saturate at small amplitudes but continue to grow into the strongly nonlinear regime [31] (see Appendix A for an example in which a small perturbation of the homogeneous steady state evolves into a large amplitude pattern in the presence of flow).

To study the long time behavior (steady state) far away from the spatially homogeneous steady state, we performed finite element simulations in Mathematica [42]. To interpret the results of these numerical simulations, we will use local equilibria theory, building on the phase-space analysis introduced in Refs. [31,37].

Figure 4A shows the space-time plot (kymograph) of a system where there is initially no flow ($t<t_{0}$), such that the system is in a stationary state with a single peak. For such a stationary steady state, diffusive fluxes on the membrane and in the cytosol have to balance exactly. This diffusive flux balance imposes the constraint that in the (m,c)-phase plane, the trajectory corresponding to the pattern lies on a straight line with slope $-D_{m}/D_{c}$, called ‘flux-balance subspace’ (FBS) [31] (see light blue line in Figure 4C). At the plateaus and inflection points of the pattern, the net diffusive flow vanishes and attachment and detachment are balanced, i.e., the system is locally in reactive equilibrium (f=0). Hence, plateaus and inflection points of the spatial concentration profile correspond to intersection points between the reactive nullcline and the FBS in the (m,c)-phase plane (blue and green points in Figure 4C). At the first intersection point (blue), the nullcline slope is larger than the FBS slope. Thus, by the slope criterion $s_{\mathrm{nc}}<-D_{m}/D_{c}$ for lateral instability, this point corresponds to a laterally stable state in the spatial domain—i.e., a plateau. Following a spatial perturbation, the concentrations will relax back towards the flat plateau.

At the second intersection point (green point in Figure 4C), the nullcline slope is more negative than the FBS slope, indicating a laterally unstable state. This state corresponds to the inflection point of the pattern and the lateral instability there can be thought of as “spanning” the interfacial region of the pattern that connects the two plateaus. An in-depth analysis of stationary patterns based on these geometric relations in phase space can be found in Ref. [31]. Here we ask how the phase portrait changes in the presence of flow.

At time $t=t_{0}$, a constant cytosolic flow in the positive *x*-direction is switched on. Consistent with the expectation from linear stability analysis, we find that the peak propagates against the flow direction in the negative *x*-direction (solid lines in Figure 4A). The diffusive fluxes no longer balance for this propagating steady state, such that the phase-space trajectory is no longer embedded in the FBS. Instead, as advective flow shifts the cytosol concentration profile relative to the membrane profile, the phase-space trajectory becomes a loop (Figure 4C). On the upstream side of the peak, the cytosolic density is increased, such that net attachment—which is proportional to the cytosolic density—is increased relative to net detachment. Conversely, the reactive balance is shifted towards detachment on the downstream side. As a result of the reactive flow is approximately proportional to the distance from the reactive nullcline in phase space, the asymmetry between net attachment and detachment on the upstream and downstream side of the peak can be estimated by the area enclosed by the loop-shaped trajectory in phase space.

To test whether the attachment–detachment asymmetry explains the propagation speed of the peak, we estimate the enclosed area in phase space by the difference in cytosolic concentrations at the points $c_{L}$ and $c_{R}$ (black dots in Figure 4C,D) where the loop intersects the reactive nullcline (f=0 black line Figure 4C). At these points, the system is in a local reactive equilibrium. Indeed, we find that the propagation speed of the pattern obtained from numerical simulations (black open squares in Figure 4B) is well approximated by the difference in cytosolic density ($v_{p}\propto c_{L}-c_{R}$) for all flow speeds (orange open circles in Figure 4B). Furthermore, in the limit of slow and fast flow, the peak propagation speed is well approximated by the propagation speed of the unstable traveling mode with the longest wavelength, as obtained from linear stability analysis (The phase velocity depends on the mode’s wavelength. The relevant length scale for the peak’s propagation is its width, which is approximately given by $2\pi /q_{\max}$ at the pattern’s inflection point [31]. Thus, we infer the peak propagation speed from −Imσ(qmax)/qmax at the inflection point of the stationary peak.). For small flow speeds, the pattern’s propagation speed $v_{p}$ increases linearly with $v_{f}$ (cf. Equation (Equation 7)) and for large flow speeds the pattern speed is proportional to $1/v_{f}$ (cf. Equation (Equation 10)).

In summary, we found that the peak propagation speed in the slow and fast flow limits is well described by the propagation speed of the linearly unstable mode with the longest wavelength (i.e., the right edge of the band of unstable modes $q_{\max}$). Moreover, we approximated the asymmetry of protein attachment by the area enclosed by the density distribution in phase space, and found that this is proportional to the peak speed for all flow speeds.

## 5. Flow-Induced Transition from Mesa to Peak Patterns

So far we have studied the propagation of patterns in response to cytosolic flow. Next, we will show how cytosolic flow can also drive the transition between qualitatively different pattern types. We distinguish two pattern types exhibited by McRD systems, peaks, and mesas [30,31]. Mesa patterns are composed of plateaus (low density and high density) connected by interfaces, while a peak can be pictured as two interfaces concatenated directly (cf. Figure 5A). Mesa patterns form if protein attachment saturates in regions of high total density, forming a plateau there. As we argued above, the low- and high-density plateaus correspond to laterally stable steady states, marked—in the phase plane—by intersection points between the FBS and the reactive nullcline where the nullcline slope is larger than the FBS slope. Peaks form if the attachment rate does not saturate at high density, i.e., if the third intersection point between nullcline and FBS is not reached [31]. Thus, while the amplitude of mesa patterns is determined by the attachment–detachment balance in the two plateaus, the amplitude (maximum concentration) of a peak is determined by the total mass available in the system [31].

How does protein transport affect whether a peak or a mesa forms? As we argued above, a peak pattern forms if protein attachment in regions of high density does not saturate. In general, this will happen if attachment to the membrane depletes proteins from the cytosol slower than lateral transport can resupply proteins (see Figure 5A). Let us first recap the situation without flow, where proteins are resupplied by diffusion from the detachment zone to the attachment zone across the pattern’s interface with width $\ell_{\mathrm{int}}$. Thus, a peak pattern forms if the rate of transport by cytosolic diffusion is faster than the attachment rate ($D_{c}/\ell_{\mathrm{int}}^{2}\gg \tau_{\mathrm{react}}^{-1}$). Further using that the interface width is given by a balance of membrane diffusion and local reactions ($\ell_{\mathrm{int}}^{2}∼\tau_{\mathrm{react}}D_{m}$), we obtain the condition $D_{c}\gg D_{m}$ for the formation of peak patterns.

In terms of phase space geometry, this means that the slope $-D_{m}/D_{c}$ of the flux-balance subspace in phase space must be sufficiently shallow. For a steep slope $-D_{m}/D_{c}$ of the FBS, the membrane concentration saturates at the point where the FBS intersects with the reactive nullcline blue dots in Figure 5A. There, attachment and detachment balance such that a mesa forms (Figure 5A). For faster cytosol diffusion, the flux-balance subspace is shallower such that the third FBS-NC intersection point shifts to higher densities. Thus, for sufficiently fast cytosol diffusion a peak forms (Figure 5B).

Adding slow cytosolic flow does not significantly contribute to the resupply of the cytosolic sink (i.e., attachment zone) and therefore does not alter the pattern type (Figure 5C). In contrast, when cytosolic protein transport (by advection and/or diffusion) is fast compared to the reaction kinetics, the cytosolic sink gets resupplied quickly, leading to a flattening of the cytosolic concentration profile. Accordingly, the density distribution in phase space approaches a horizontal line, both for fast cytosolic diffusion (Figure 5B) and for fast cytosolic flow (Figure 5D). As a consequence, the point where the density distribution meets the nullcline shifts towards larger membrane concentrations, resulting in an increasing amplitude of the mesa pattern. Eventually, when the amplitude of the pattern can not grow any further due to limiting total mass, a peak pattern forms (Figure 5B,D). Hence, an increased flow velocity can cause a transition from a mesa pattern to a peak pattern (see Appendix A).

In summary, we found that cytosolic flow can qualitatively change the membrane-bound protein pattern from a small-amplitude, wide mesa pattern to a large-amplitude, narrow peak pattern. In cells, such flows could therefore promote the precise positioning of polarity patterns on the membrane. Furthermore, we hypothesize that flow can contribute to the selection of a single peak by accelerating the coarsening dynamics of the pattern via two distinct mechanisms. First, flow accelerates protein transport that drives coarsening. Second, as peak patterns coarsen faster than mesa patterns [30,43], flow can accelerate coarsening via the flow-driven mesa-to-peak transition. Such fast coarsening may be important for the selection of a single polarity axis, e.g., a single budding site in *S. cerevisiae* [4], for axon formation in neurons [44], and to establish a distinct front and back in motile cells [2,45].

## 6. Flow-Induced Pattern Formation

So far we have studied how a uniform flow profile affects pattern formation on a domain with periodic boundary conditions, representing circular flows along the cell membrane and bulk flows in microfluidic in vitro setups. However, flows in the vicinity of the membrane can be non-uniform. For example, one (or more) components of the pattern forming system may be embedded in the cell cortex [13,15,46] which is a contractile medium driven by myosin-motor activity. Furthermore, the incompressible cytosol can flow in the direction normal to the membrane, such that the 3D flow field of the cytosol is perceived as a compressible flow along the membrane [9]. In this section, we will discuss how such non-uniform, uni-directional flows lead to pattern formation.

A non-uniform flow transports the proteins at different speeds along the membrane. Starting from a spatially homogeneous initial state, such a non-uniform flow leads to a redistribution of mass. It has been demonstrated in previous work that this non-uniform flow can induce pattern formation even if the homogeneous steady state is laterally stable (i.e., there is no spontaneous pattern formation) [7,13,15]. Based on numerical simulations, a transition from flow-guided to self-organized dynamics has been reported [15]. However, the physical mechanism underlying this transition, and what determines the transition point have remained unclear.

We address this question using the two-component model, which serves a conceptual model that mimics the qualitative behavior of the more complex PAR system [33]. While flow in the PAR system is governed by the myosin concentration, we assume a stationary parabolic flow profile that vanishes at the system boundaries (Figure 6A, top). We use a one-dimensional domain with no-flux boundary conditions that correspond to the symmetry axis of a rotationally symmetric flow profile. In the following, we describe the flow-induced dynamics starting from a spatially homogeneous steady state to the final polarity pattern observed in numerical simulations (see Appendix A). Figure 6 visualizes these dynamics in real space (A) and in the (m,c)-phase plane (B). To relate our findings to the previous study Ref. [15], we also visualize the dynamics in an abstract representation of the state space (comprising all concentration profiles) used in this previous study. In this state space, steady states are points and the time evolution of the system is a trajectory (thick blue/orange line in Figure 6C).

Starting from the homogeneous steady state (*i*), the non-uniform advective flow redistributes mass in the cytosol (*ii*). Due to this redistribution of mass, the local reactive equilibria shift as we have seen repeatedly here and in earlier studies of mass-conserving systems [31,47]. In fact, as long as the gradients of both the membrane and cytosol profiles are shallow, the concentrations remain close to the local equilibria, as evidenced by the density distribution in phase space spreading along the reactive nullcline (see profile (*ii*) in Figure 6A,B). As long as there is no laterally unstable region, the mass accumulation is limited by the counteracting diffusive flow in the cytosol. Eventually, the region where mass accumulates (here the right edge of the domain) enters the laterally unstable regime (see profile *iii*). In this laterally unstable region, cytosol diffusion will enhance the accumulation of mass via the mass-redistribution instability, until it is limited by the much slower membrane diffusion. In the phase plane (Figure 6B), the laterally unstable regime corresponds to the range of total densities $\bar{n}$ where the nullcline slope has a steeper negative value than the flux-balance subspace slope ($s_{\mathrm{nc}}<-D_{m}/D_{c}$) (More precisely, the size of the laterally unstable region must be larger than the shortest unstable mode (corresponding to the right edge of the band of unstable modes in the dispersion relation (Figure 2A))). The mass-redistribution instability in this region, based on the self-organized formation of attachment and detachment zones (cf. Section 3.2) will lead to the formation of a polarity pattern there (*iv*). Thus, the onset of a regional lateral instability marks the transition from flow-guided dynamics to self-organized dynamics.

In the abstract state space visualization (Figure 6C) the area shaded in orange indicates the polarity pattern’s basin of attraction comprising all states (concentration profiles) where a spatial region in the system is laterally unstable. In the absence of flow, states that do not exhibit such a laterally unstable region return to the homogeneous steady state (thin gray lines). Non-uniform cytosolic flow induces mass-redistribution, that can drive an initially homogeneous system (*i*) into the polarity pattern’s basin of attraction. From there on, self-organized pattern formation takes over, leading to the formation of a polarity pattern (*iv*), essentially independently of the advective flow (orange trajectory). In future work, it would be interesting to make the abstract state space representation, Figure 6C, more quantitative. For example, one could try to estimate the minimal flow velocity required to drive the system past the separatrix, i.e., into the basin of attraction of the polarity pattern. A promising approach is to use the fact that prior to the onset of regional lateral instability, the concentrations are slaved to the local equilibria that depend on the local total density. Thus, one can obtain an approximate, closed equation for the flow-driven evolution of the total density, similar to the “adiabatic scaffolding approximation” made in [31]. Solving this equation would provide a criterion for when the total density exceeds the critical density for lateral instability in some spatial region that initiates the self-organized formation of a polarity pattern there.

Similar pattern forming mechanisms based on a regional instability have previously been shown to also underlie stimulus-induced pattern formation following a sufficiently strong initial perturbation [31] and peak formation at a domain edge where the reaction kinetics abruptly change [36]. Thus, an overarching principle for stimulus-induced pattern formation emerges: To trigger (polarity) pattern formation, the stimulus, be it advective flow or heterogeneous reaction kinetics, has to redistribute protein mass in a way such that a regional (lateral) instability is triggered.

It remains to be discussed what happens once the cytoplasmic flow is switched off after the polarity pattern has formed. In general, the polarity pattern will persist (see Appendix A), since it is maintained by self-organized attachment and detachment zones, largely independent of the flow. However, as long as there is flow, the average mass on the right hand side of the system (downstream of the flow) is higher than on the left hand side. Hence, flow can maintain a polarity pattern even if the average mass in the system as a whole is too low to sustain polarity patterns in the absence of flow (see bifurcation analysis in Ref. [31]). If this is the case, the peak disappears once the flow is switched off (see Appendix A).

In summary, the redistribution of the protein mass is key to induce (polarity) pattern formation starting from a stable homogeneous state.

## 7. Conclusions and Outlook

Inside cells, proteins are transported via diffusion and fluid flows, which, in combination with reactions, can lead to the formation of protein patterns on the cell membrane. To characterize the role fluid flows play in pattern formation, we studied the effect of flow on the formation of a polarity pattern, using a generic two-component model. We found that flow leads to propagation of the polarity pattern against the flow direction with a speed that is maximal for intermediate flow speeds, i.e., when the rate of advective transport is comparable to either the reaction rates or to the rate diffusive transport in the cytosol. Using a phase-space analysis, we showed that the propagation of the pattern is driven by an asymmetric influx of protein mass to a self-organized protein-attachment zone. As a consequence, attachment is stronger on the upstream side of the pattern compared to the downstream side, leading to upstream propagation of the membrane bound pattern. Furthermore, we have shown that flow can qualitatively change the pattern from a wide mesa pattern (connecting two plateaus) to a narrow peak pattern. Finally, we have presented a phase-space analysis to elucidate the interplay between flow-guided dynamics and self-organized pattern formation. This interplay was previously studied numerically in the context of PAR-protein polarization [13,15]. Our analysis reveals the underlying cause for the transition from flow-guided to self-organized dynamics: the regional onset of a mass-redistribution instability.

We discussed implications of our results and links to earlier literature at the end of each section. Here, we conclude with a brief outlook. We expect that the insights obtained from the minimal two-component model studied here generalize to systems with more components and multiple protein species. For example, in vitro studies of the reconstituted MinDE system of *E. coli* show that MinD and MinE spontaneously form dynamic membrane-bound patterns, including spiral waves [48] and quasi-stationary patterns [49]. These patterns emerge from the competition of MinD self-recruitment and MinE-mediated detachment of MinD [50,51]. In the presence of a bulk flow, the traveling waves were found to propagate upstream [16]. Our analysis based on a simple conceptual model suggests that this upstream propagation is caused by a larger influx of the self-recruiting MinD on the upstream flanks compared to the downstream flanks of the traveling waves. However, the bulk flow also increases the resupply of MinE on the upstream flanks. As MinE mediates the detachment of MinD and therefore effectively antagonizes MinD’s self-recruitment, this may drive the membrane-bound patterns to propagate downstream instead of upstream. Which one of the two processes dominates—MinD-induced upstream propagation or MinE-induced downstream propagation—likely depends on the details of their interactions. This interplay will be the subject of future work.

A different route of generalization is to consider advective flows that depend on the protein concentrations. In cells, such coupling arises, for instance, from myosin-driven cortex contractions [15,52] and shape deformations [8,25]. Myosin-motors, in turn, may be advected by the flow and their activity is controlled by signaling proteins such as GTPases and kinases [53]. This can give rise to feedback loops between flow and protein patterns. Previous studies show that such feedback loops can give rise to mechano-chemical instabilities [54], drive pulsatile (standing-wave) patterns [55,56] or cause the breakup of traveling waves [57]. We expect that our analysis based on phase-space geometry can provide insight into the mechanisms underlying these phenomena.

## Figures and Tables

**Figure 1 cells-09-01524-f001:**
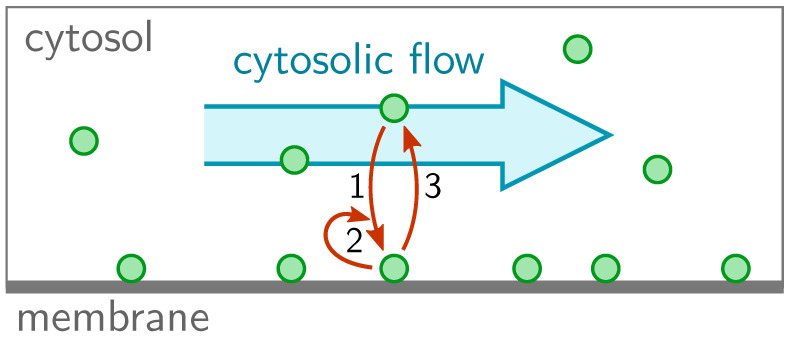
One-dimensional two-component system with cytosolic flow into the positive *x*-direction. The reaction kinetics include (**1**) attachment, (**2**) self-recruitment, and (**3**) enzyme-driven detachment.

**Figure 2 cells-09-01524-f002:**
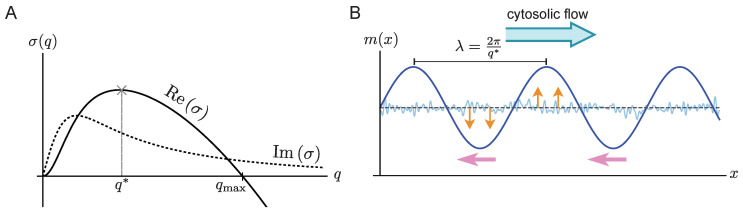
(**A**) Sketch of real (solid) and imaginary (dotted) part of a typical dispersion relation with a band $\left[0,q_{\max}\right]$ of unstable modes. (**B**) The initial dynamics of a spatially homogeneous state with a small random perturbation (blue thin line). The direction of cytosolic flow is indicated by a blue arrow. The typical wavelength (λ) of the initial pattern is determined by the fastest growing mode $q^{*}$ and the phase velocity is determined by the value of the imaginary part of dispersion relation at the fastest growing mode (vphase=−Imσ(q*)/q*). The growth of the pattern is indicated by orange arrows, while the traveling direction is indicated by pink arrows.

**Figure 3 cells-09-01524-f003:**
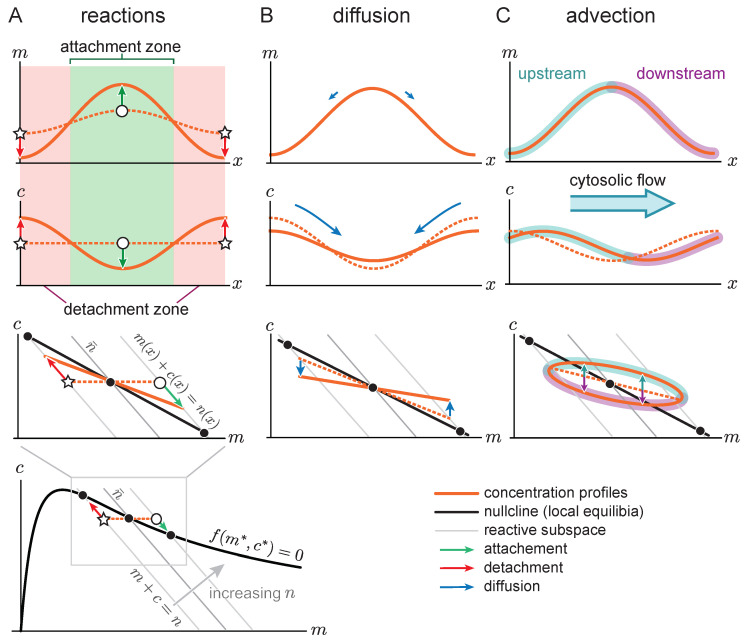
Sketch of the initial dynamics of an laterally unstable spatially homogeneous steady state. The role of reactions (**A**), diffusion (**B**), and advection (**C**) for a mass-redistribution instability are presented for the membrane (top) and cytosolic (middle) concentration profiles and in phase space (bottom). (**A**) A small perturbation of the spatially homogeneous membrane concentration (orange dashed lines in top panel) leads to a spatially varying local total density n(x), with a larger total density at the maximum of the membrane profile (open circle) and a smaller total density at the minimum (open star). These local variations in total density lead to attachment zones (green region) and detachment zones (red region). The reactive flow, indicated by the red and green arrows, points along the reactive subspace (gray lines) in phase space towards the shifted local equilibria (black circles). These reactive flows lead to the solid orange density profiles after a small amount of time. (**B**) Faster diffusion in the cytosol compared to the membrane (indicated by the large and small blue arrows in the middle and top panel, respectively), lead to net mass transport from the detachment zone to the attachment zone. Again, dashed and solid lines indicate the state before and after a short time interval of diffusive transport. (**C**) Cytosolic flow shifts the cytosolic concentration with respect to the membrane concentration (orange dashed to orange solid lines), increasing the cytosolic concentration on the upstream side of the pattern and decreasing the cytosolic concentration on the downstream side. In phase space, the trajectory of this density profile forms a ‘loop’.

**Figure 4 cells-09-01524-f004:**
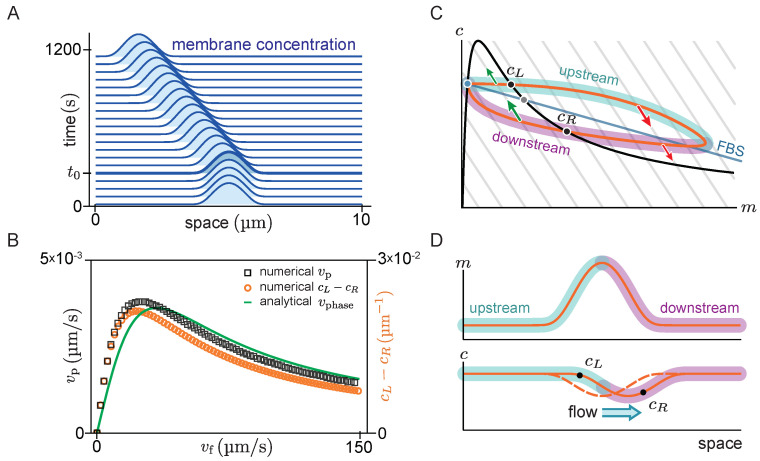
Pattern dynamics far from the spatially homogeneous steady state. (**A**) Time evolution of the membrane-bound protein concentration. At time $t_{0}=240\;s$ a constant cytosolic flow with velocity $v_{f}=20\;\mu m/s$ towards the right is switched on (cf. Movie 3). (**B**) Relation between the peak speed ($v_{p}$) and flow speed ($v_{f}$). Results from finite element simulations (black open squares) are compared to the phase velocity of the mode $q_{\max}$ obtained from linear stability analysis (green solid line) and to an approximation (orange open circles) of the area enclosed by the density distribution trajectory in phase space (area enclosed by the ‘loop’ in **D**). (The domain size, L=10 μm, is chosen large enough compared to the peak width such that boundary effects are negligible.) (**C**) A schematic of the phase portrait corresponding to the pattern in **D**. The density distribution in the absence of flow is embedded in the flux-balance subspace (FBS) (blue straight line). In the presence of flow, the density distribution trajectory forms a ‘loop’ in phase space. The upstream and downstream side of the pattern are highlighted in cyan and magenta, respectively. Red and green arrows indicate the direction of the reactive flow in the attachment and detachment zones, respectively. At intersection points of the density distribution with the nullcline ($c_{L}$ and $c_{R}$) the system is at its local reactive equilibrium. (**D**) Sketch of the membrane (orange solid line, top) and cytosolic (orange dashed line, bottom) concentration profiles for a stationary pattern in the absence of cytosolic flow. Flow shifts the cytosol profile downstream (orange solid line, bottom).

**Figure 5 cells-09-01524-f005:**
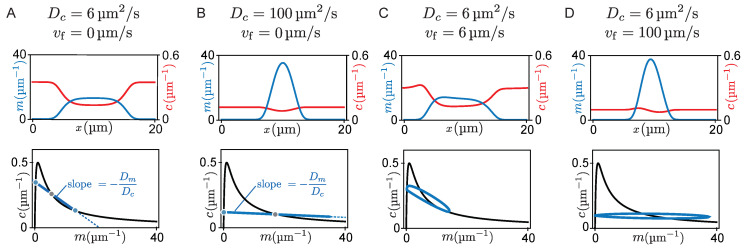
Demonstration of the transition from a mesa pattern to a peak pattern. Each panel shows a snapshot from finite element simulations in steady state. Top concentration profiles in real space; bottom: corresponding trajectory (blue solid line) in phase space. (**A**) Mesa pattern in the case of slow cytosol diffusion and no flow. The two plateaus (blue dots) and the inflection point (gray dot) of the pattern correspond to the intersection points of the FBS (blue dashed line) with the reactive nullcline (black line). (**B**) For fast cytosol diffusion, the third intersection point between FBS and nullcline lies at much higher membrane concentration such that it no longer limits the pattern amplitude. Therefore, a peak forms whose amplitude is limited by the total protein mass in the system. (**C**) Slow flow only slightly deforms the mesa pattern, compare to (**A**). Fast cytosolic flow leads to formation of a peak pattern (**D**), similarly to fast diffusion. Parameters: $\bar{n}=7\;\mu m^{-1},D_{m}=0.1\;\mu m^{2}/s$, and L=20 μm.

**Figure 6 cells-09-01524-f006:**
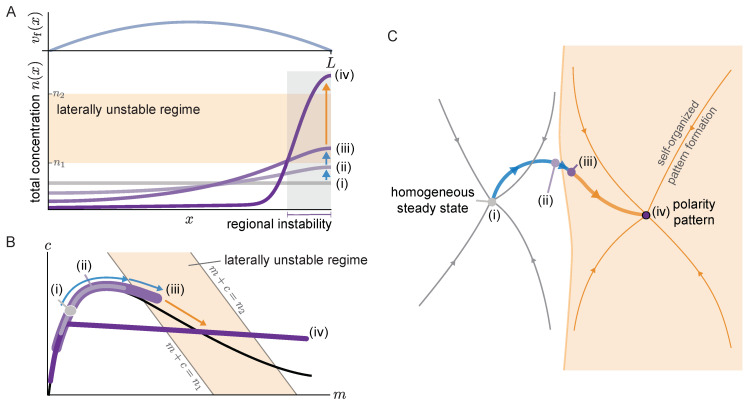
Flow-driven protein mass accumulation can induce pattern formation by triggering a regional lateral instability. (**A**) Top: quadratic flow velocity profile: $v_{f}\left(x\right)/v_{\max}=1-4{\left(x/L-1/2\right)}^{2}$. Bottom: illustration of the total density profiles at different time points starting from a homogeneous steady state (*i*) to the final pattern (*iv*); see Movie 5. Mass redistribution due to the non-uniform flow velocity drives mass towards the right hand side of the system, as indicated by the blue arrows. The range of total densities shaded in orange indicates the laterally unstable regime determined by linear stability analysis. Once the total density reaches this regime locally, a regional lateral instability is triggered resulting in the self-organized formation of a peak (orange arrow). (**B**) Sketch of the phase space representation corresponding to the profiles shown in A. Note that the concentrations are slaved to the reactive nullcline (black line) until the regional lateral instability is triggered. (**C**) Schematic representation of the state space of concentration patterns in a case where both the homogeneous steady state and a stationary polarity pattern are stable. Thin trajectories indicate the dynamics in the absence of flow and the pattern’s basin of attraction is shaded in orange. The thick trajectory connecting both steady states shows the flow-induced dynamics, corresponding to the sequence of states (*i*)–(*iv*) shown in **A** and **B**.

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
