# Peer review of "Flow Induced Symmetry Breaking in a Conceptual Polarity Model"

_cells, 2020, doi:10.3390/cells9061524_

Round 1

Reviewer 1 Report

The manuscript describes pattern formation in a 2-component mass-conserving reaction diffusion (2cMcRD) system under the influence of flow in one of the compartments. As this model is applied to cell polarity, the two compartments correspond to the cytosol and the plasma membrane, respectively, and the flow occurs in the cytosol. The authors find that counterintuitively the pattern moves upstream, i.e. against the direction of the flow. Moreover, Wigbers et al. demonstrate that the maximal speed of the moving pattern is attained at intermediate flow velocities, and that cytosolic flows can change mesa patterns into peak patterns. The manuscript is extremely well written and provides physical explanations for the pattern formation in addition to mathematical arguments. Overall, this is an excellent study that addresses a critical question in cell biology. I have a few comments that I would like the authors to address before the manuscript can be accepted for publication.

- My main comment relates to Section 6. Compared to the other result sections, I found it less clear both in terms of the setup and the explanations. In contrast to the periodic boundary conditions used in the other sections, the 2cMcRD system is now modelled subject to no-flux boundary conditions. While I understand the use of periodic boundary conditions (as circular flux along the membrane), it would be good to provide a biophysical explanations for the no-flux boundary conditions. Moreover, given a uni-directional flux (albeit with space-varying amplitude) of sufficient strength, would the no-flux boundary conditions not always result in a one-sided accumulation of the total concentration n and hence a peak of the membrane pattern at that side, since mass is transported towards the boundary and kept there?

- The authors explicitly mention a stationary parabolic flow profile. It is not clear from the presentation how this shape enters the results. Also, do the results depend on the actual shape of the flow profile? If so, it would be good to show some simulations for this. Along similar lines, could I ask the authors do speculate how patterns could change if the flow is modulated in a spatio-temporal fashion?

- As I understand Figure 6C was included to make contact with a previous study (Ref 11). As it currently stands, I do not think that it adds much to the manuscript as it is too much of a schematic. However, if the figure were made more quantitative, it would be a great addition to see the actual shape of the connection between the two different linearly stable fixed points and the position of the separatrix. I appreciate that this might involve quite some work, so I leave it as a suggestion.

Here are some minor comments.

- I would formally introduce the vector u on page 4 as u(x,t)=(c(x,t),m(x,t)) to provide information about the ordering of the components in u.

- Could the authors provide more detail on why the condition in line 162 entails that the instability criterion becomes independent of the the diffusion coefficients? I can see that one could drop the 1 in the bracket, but then how does this lead to the above independence?

- It might be worth adding some lines to explain why patterns are bounded by the intersections between the reactive nullcline and the FBS (e.g. when discussing Figure 5). This might also help making the paragraph starting on line 290 clearer.

- The manuscript states that the non-uniform velocity in Section 6 vanishes at the boundaries. However, the expression provided in the caption of Figure 6 does not vanish at the boundaries. It has the right shape, i.e. a maximum at L/2, but non-vanishing values at the boundaries.

Some typos.

- l.104: Additional full stop in front of Ref [27]. Also, should it be σ(q)<0 instead of σ(0)<0?

- l.171: Should there be a minus sign in front of Im σ?

- l.206: additional "where"

- l.240: "corresponds inflection points" -> "corresponds to inflection points"

- l.276: "cf. Fig. 4" -> "cf. Fig. 5"?

- l.343: "has a steeper negative than" -> "has a steeper negative value than"

Reviewer 2 Report

Here, the authors study the travelling wave patterns that emerge in a protein that can bind a membrane or cell cortex, where it only diffuses, and be advected in cytosol, by a cytosolic streaming velocity that is set by processes which are not discussed here. This is a reframing of an instability that is known and has been studied in other contexts, such as the growth of vegetation to a cell biological context, where to my knowledge this phenomenon has not been discussed before.

The paper is very well written and technically correct. The relevant literature is referenced. Maybe a little more attention could be given to disentangle this system from the more complex scenario studied in the context of the PAR system, for instance in C. elegans. The key differences (that flows are driven by the protein itself, that advection in the cortex is important) could be stated and discussed.

In summary, this paper reframes known physics to the cell biological context. It does so without technical mistakes, and matters are well explained. It is entirely appropriate for publication in cells as is.
